# Ergonomic Risk Assessment of Dental Students—RULA Applied to Objective Kinematic Data

**DOI:** 10.3390/ijerph181910550

**Published:** 2021-10-08

**Authors:** Kim Sarah Blume, Fabian Holzgreve, Laura Fraeulin, Christina Erbe, Werner Betz, Eileen M. Wanke, Doerthe Brueggmann, Albert Nienhaus, Christian Maurer-Grubinger, David A. Groneberg, Daniela Ohlendorf

**Affiliations:** 1Institute of Occupational Medicine, Social Medicine and Environmental Medicine, Goethe University, 60596 Frankfurt am Main, Germany; kim.blume@gmx.de (K.S.B.); holzgreve@med.uni-frankfurt.de (F.H.); maltry@med.uni-frankfurt.de (L.F.); wanke@med.uni-frankfurt.de (E.M.W.); brueggmann@med.uni-frankfurt.de (D.B.); Christian.maurer.cm@gmail.com (C.M.-G.); arbsozmed@uni-frankfurt.de (D.A.G.); 2Department of Orthodontics, School of Dentistry, University Medical Centre of the Johannes Gutenberg University, 55131 Mainz, Germany; erbe@uni-mainz.de; 3Institute of Dentistry, Goethe-University, 60596 Frankfurt am Main, Germany; w.betz@em.uni-frankfurt.de; 4Principles of Prevention and Rehabilitation Department (GPR), Institute for Statutory Accident Insurance and Prevention in the Health and Welfare Services (BGW), 20251 Hamburg, Germany; albert.nienhaus@bgw-online.de

**Keywords:** ergonomic risk, dental students, RULA, kinematic analysis, dental activities, MSD

## Abstract

Musculoskeletal disorder (MSD) is already prevalent in dental students despite their young age and the short duration of dental practice. The current findings state that the causes of MSD are related to posture during dental work. This study aims to investigate the ergonomic risk of dental students. In order to analyze the ergonomic risk of dental students, 3D motion analyses were performed with inertial sensors during the performance of standardized dental activities. For this purpose, 15 dental students and 15 dental assistant trainees (all right-handed) were measured in a team. Data were analyzed using the Rapid Upper Limb Assessment (RULA), which was modified to evaluate objective data. Ergonomic risk was found for the following body parts in descending order: left wrist, right wrist, neck, trunk, left lower arm, right lower arm, right upper arm, left upper arm. All relevant body parts, taken together, exhibited a posture with the highest RULA score that could be achieved (median Final Overall = 7), with body parts in the very highest RULA score of 7 for almost 80% of the treatment time. Dental students work with poor posture over a long period of time, exposing them to high ergonomic risk. Therefore, it seems necessary that more attention should be paid to theoretical and practical ergonomics in dental school.

## 1. Background

Musculoskeletal disorders (MSD) and the resulting pain not only have a high prevalence among dentists, but these physical impairments are also evident in dental students [1,2,3,4,5]. It alarming that even young dental students, who have only been practicing dentistry for a short period of time, suffer from MSD and pain. Seventy-five years ago, in 1946, Biller [6] showed that 65% of dentists complained about back pain. The evidence that even dental students are affected by MSD [7,8,9,10,11] has become visible via the use of a modified Nordic questionnaire. By using this questionnaire (derived from the Standardized Nordic Questionnaire), the prevalence of MSD showed that 85% of the dental students reported MSD in at least one body region [12]. A high prevalence of MSD in the upper body regions has also been reported for Australian dental students (between 64% and 93% with MSD) [13]. A similar prevalence has been demonstrated worldwide [7,9,14,15,16].

MSD can become chronic and account for a large proportion (40%) of all chronic diseases [17,18]. Besides the health consequences for ill dental students/dentists, the economic burden of MSD has been proven to be severe [19]. MSD has also been shown to have a negative impact on work motivation and the quality of dental work [8]. MSD also increase the risk of sick leave, work disability and early career exit in dentistry [17,19,20,21,22,23,24,25]. Thus, in order to minimize the harmful effects of MSD in dentistry, it is important to identify the risk factors. The main risk factors for developing musculoskeletal conditions in dentistry are using vibrating instruments, the patient’s mouth being a small, difficult-to-see work area, the difficult positioning of the patients, the precise and repetitive movements in a confined space and the near-static posture adopted for long periods of time (longer than four seconds) [26,27,28]. Researchers [29] found that musculoskeletal pain reported by dental students positively correlated with poor posture. Studies [13,30,31] have demonstrated that the head and trunk adopt a static posture for 27.4% and 23.6% of the treatment time during dental work, respectively. Additionally, static positions are often riskier for the musculoskeletal system than dynamic movements in dentistry. Neves et al. [32] found a significant correlation between the students’ difficulties in preclinical restorative procedures and their difficulties in maintaining an ergonomic posture. The results of a study by Diaz-Gaballero et al. [7] showed that exaggerated body flexion or cervical torsion, performed to improve the view of the oral cavity, caused musculoskeletal pain among Colombian dental students.

By using a validated posture assessment instrument, “Branson’s dental operator posture assessment instrument” (PAI), researchers found a strong prevalence of postural problems among dental students [12]. In 2020, a study analyzed the students’ muscle activity during dental work using electromyography and recorded the spinal tilt via an inclinometer; this revealed a high ergonomic risk during dental work due to back flexion of more than 20 degrees and unfavorable muscle activity of the neck muscles [33]. Furthermore, by using the Rapid Upper Limb Assessment (RULA), a study of dental students found that they were required to maintain a posture that can lead to muscle fatigue and pain [8]. Another study also used the RULA tool to identify dental students and the resulting moderate to high enveloping MSD [34]. It is also alarming to discover that 80.8% of dental students are not aware of the ergonomically correct postures during dental work [31]. Cervera-Espert et al. [35] interviewed dental students and found that the majority of students were not familiar with ergonomics and, accordingly, did not sit correctly in the dentist’s chair. Furthermore, neither knowledge of the ergonomic requirements nor their practical application among dental students were found to be satisfactory in a survey by Garbin et al. [36]. Regarding the postural awareness of dental students, observations and interviews have established that their knowledge can be improved and that better awareness of proper posture during patient care can help to reduce the risk of MSD [37]. In addition, Faust et al. [38] demonstrated that the hands-on teaching of trained dental educators positively influences adherence to ergonomic principles. Three months after ergonomic training, 49% of the participating dental students reported a reduction in musculoskeletal pain [39]. The main reasons given for not adopting ergonomic postures are lack of attention and practice, and forgetfulness (44.8%). Difficulties in visualizing the operator panel or the procedure being performed (27.6%) are also mentioned [40].

Various international studies [10,12,13,20,41] have demonstrated a high prevalence of MSD and musculoskeletal pain among dentists and dental students worldwide; however, the data of these studies were mainly based on surveys. Given the diversity of the reported outcomes and the lack of kinematic analyses, further research is needed to elucidate further the risk and impact of posture among dental students.

There are many MSD assessment methods such as PAI, Rapid Entire Body Assessment (REBA), Key Indicator Index (KIM), Ovako Working Posture Assessment System (OWAS) or RULA. In the present study, a combination of Rapid Upper Limb Assessment, forming one ergonomic risk assessment, and inertial motion capture technology has been applied to quantify the dental students’s posture and the ergonomic risk; this is an automated process based on kinematic data that evaluate the ergonomic risk of students during dental activities using a dummy head. RULA [42] was used as an evaluation method because it has a good representation of the aforementioned body regions. In addition, RULA [42] is an international, frequently used observation method for classifying the ergonomic risks of work processes. The exact analysis procedure can be found in Maurer-Grubinger et al. [43]. This resulted in the objective of quantifying the ergonomic risk based on RULA. This present analysis of the activities of the dental students is part of the SOPEZ project [44] which investigates the ergonomic risk of dentists and dental assistants working in different dental treatment concepts.

## 2. Material and Methods

### 2.1. Subjects

Fifteen teams were formed, with each consisting of one dental student and one dental assistant trainee, all of whom were right-handed in dental work. The personal data of the participants can be found in Table 1.

The inclusion criterion was that the students already had practical experience in dentistry to fulfill the requirements of performing dental activities. Dental assistant trainees were eligible to participate in the study from their first year of training.

Exclusion criteria included acute musculoskeletal injuries, rheumatism, limiting spinal deformities or stiffness of the spine and genetic muscular diseases. Another requisite was that if the subjects had previously undergone surgery, then this should have taken place more than 2 years previously.

In this study, the focus was on the 15 dental student participants and only their measurement data were evaluated and analyzed here. The measurement as a treatment team was, nevertheless, important, in order to recreate the treatment situation as realistically as possible.

This study was approved by the Ethics Committee of the Department of Medicine at Goethe University Frankfurt am Main (356/17). All experiments were completed in accordance with the relevant guidelines and regulations, and all participants provided written informed consent.

### 2.2. Measurement System

The recording of the body postures was performed using the MVN inertial motion capture system from Xsens (Enschede, Netherlands). This motion analysis system captures information on acceleration, velocity, joint angles and the position parameters of the human body using 17 sensors. The sampling rate of the system is 240 Hz. The continuous data were “downframed” to 24 Hz since no highly dynamic movements were recorded. We obtained 24 exposures per second. There were 24 recordings stored every second. At the end of each activity, the mean value was calculated from the sum of each measurement recording. The measurement error is reported by the manufacturer to be ±1%.

A total of 17 sensors were attached to the Xsens full-body suit worn by the subject, and the kinematic parameters of 22 joints and 3 degrees of freedom were calculated.

After dressing in the measurement suit and attaching the motion sensors, the subjects were calibrated according to the manufacturer’s specifications and only the calibration quality of “good” was used. The recording was performed in the “no level” function; this setting is recommended by the manufacturer if the recordings are not performed on a level floor (the subjects were sitting on a dentist chair). Thus, the hip segment formed the reference point of the individual´s coordinate system. Finally, the “HD reprocessing” filter was applied to all recordings. According to the manufacturer, this provides the optimum data quality and is included in the MVN Analyze software.

### 2.3. Measurement Protocol

After calibration, the dental activities were performed on a dummy head. The measurements were carried out according to dental treatment concept 1 (Figure 1) using a saddle chair. Since only dental treatment concept 1 is taught at German universities, students are only familiar with this concept. Figure 2 illustrates the study procedure and shows 2 study participants with the measuring suit in which all sensors are integrated. The following dental activities were to be performed by each student in a standardized sequence (Table 2):

All tasks in the four dental quadrants were performed using treatment concept 1, defined by Ohlendorf et al. [44]. A quadrant is understood to be one half of the jaw. Quadrant 1 is the right half of the upper jaw. Quadrant 2 is the left half of the upper jaw. Quadrant 3 is the left half of the lower jaw. Quadrant 4 is the right half of the lower jaw. Tasks mean the different dental activities. A dental treatment concept refers to the different arrangement of the dental chair, dental tray and dental instruments during a treatment. Based on the individual skills of the students, different amounts of time were needed to complete each task. The dental tasks in the first to third quadrants took 4–5 min, while removal tartar in the fourth quadrant took the least time, at around 3 min. As all students were right-handed, the contra-angle handpiece was mostly held in the right hand in the first to third quadrants. In the fourth quadrant, the scaler was held in the right hand throughout the task. The students could use the mirror in their left hand according to their individual habits. Further hand tools are mentioned in Table 2. The setting of the dental lamp had the same standardized position, placed exactly and vertically above the dummy head at the beginning of each measurement. The light setting was allowed to be varied by the subject during the measurement. In contrast, no variation in the dummy head position was allowed during the measurement. The subject had to adjust the phantom head position to the respective quadrant before the measurement. The treatment chair had the same, mostly horizontal, position at all times of the measurement and also in each quadrant, and was not allowed to be changed, in accordance with Ohlendorf et al. [44].

At the beginning and end of each measurement, a start/end position (the left and right hand resting on the respective thigh and the face and gaze directed towards the dummy head) was adopted to represent the beginning and end of the measurement.

The entire measurement process was filmed in an all-round view (iPad Air) to precisely assign the movements in retrospect and to investigate any deviations, if possible. To ensure that the measurement recordings and camera recordings coincided in time, they were synchronized using the MVN Analyze 2020.2 software (Xsens, Enschede, The Netherlands).

### 2.4. RULA

To evaluate the quantified working posture from an ergonomic perspective, RULA was applied to the data. The kinematic analysis refers to body regions such as the neck, shoulders, trunk, arms and hands. Using predetermined representations of different postures, an overall “final” posture can be quantified [45,46]. For the ergonomic assessment of the posture, a risk score was compiled from the data for each captured image point using RULA. Based on this, a median score and IQR for the wrist, upper arm and forearm (Section A), and the neck and trunk (Section B), could also be determined. The evaluation protocol consisted of three evaluation scores. The wrist and arm score involved the measurements of the upper arm, forearm and wrist, while the neck, trunk and leg score included the measurements of the neck, trunk and legs. Subsequently, the values for the static muscle work and strength were added to the determined posture scores to form the two mentioned scores, resulting in the “wrist and arm score” and the “neck, trunk, leg score”, respectively. Then, the two averages of the wrist and arm scores and the neck, trunk, and leg scores were added together, resulting in a final score. This final risk score was calculated based on the RULA method (McAtamney and Corlett, 1993) [45] and indicates the measure of the MSD risk of the activity and quadrant under investigation. This total RULA score ranged from one to seven for the surveyed posture and the associated ergonomic risk rated according to the following classifications [47]:-1–2: Posture is acceptable if not maintained.-3–4: Further investigation needed. May need changes.-5–6: Further investigation and changes needed soon.-7: Investigation and changes required immediately.

For the calculation of the RULA, adjustments to RULA, specified in Table 3 were necessary to apply it to the quantitative data of the IMU sensors.

Table 3 shows the necessary adjustments to the original RULA, since all RULA thresholds must be quantitatively defined in the present approach [47,48]. The combination of RULA and kinematic data enables a differentiated evaluation of the ergonomic risk based on three outcomes, relative to both total RULA score and individual, body-region RULA scores:Median + interquartile distance (IQR);Relative time score;Ergonomic risk potential (ERP).

The *relative time score* was calculated from the relative portion of the time spent at each RULA score. The following formula was used:relative time spent at RULA score 1×1 + relative time spent at RULA score 2×2 + relative time spent at RULA score 3×3(..) + relative time spent at RULA score 7×7(1)

The resulting ERP is the proportion of the *relative time score* to the maximum number of RULA scores that can be achieved. This is important, especially if the ergonomic risks of the local scores are to be compared; since different maximum RULA scores can be achieved, the higher the value of the *relative time score* and ERP, the higher the ergonomic risk of the individual body regions.

Besides the total RULA score, the quantitative approach enables the calculation of ergonomic risk for different body regions. These parameters are termed “local scores” and are listed as follows:Neck Score                                            -      RULA Step 9Trunk Score                                          -      RULA Step 10Upper Arm Score  (left and right)     -      RULA Step 1Lower Arm Score  (left and right)     -      RULA Step 2Wrist Score             (left and right)     -      RULA Step 3 + 4

### 2.5. Statistical Analysis

The statistical data analysis was performed in Microsoft Excel 2016 and Matlab R2020a (The Mathworks Inc., Natick, MA, USA). In addition, all important inertial measurement unit (IMU) data were analyzed in Matlab and exported as Excel files. The Kolmogorov—Smirnov—Lilliefors test was used to test the data for normal distribution. However, since the data were non-normally distributed, the median and IQD were determined for descriptive statistics.

## 3. Results

Relative time score, ERP, and the original risk classification are correlated, with:

ERP directly calculated from the relative time score: rho = 1, *p* < 0.0001,

Relative time score, original risk classification: rho = 0.59, *p* = 0.0232

ERP, original risk classification: rho = 0.59, *p* = 0.0232

This underlines the validity of the relative time score and the ERP compared to the original risk score.

Table 4 includes the median + IQD and the *relative time score* of all assessed RULA steps of the different body regions, as well as the final score. The maximum score achieved by each respective body part is also given and shows the ergonomic risk is in each case. Table 4 also contains the *relative time scores* of all assessed RULA values of the different body regions, showing the ergonomic risks of the different measured body parts. In summary, the evaluation of the *relative time scores* shows that the wrists are the most affected, with the left side being the most at risk. Subsequently, the neck and trunk are moderately at risk, followed by the right and left lower arm, while the right and left upper arm have a rather low ergonomic risk.

Table 4 also contains the ERP of the different measured body parts, from the highest to the lowest risk, with the following order: left lower arm, left wrist, right wrist, right lower arm, neck, trunk, right upper arm, left upper arm. Thus, the ERP shows that the left lower arm, followed by the left lower wrist, has the highest ergonomic risk of all the measured body parts. Similarly, both the ERP and the median show that the left side of the wrists and lower arms are more affected, and that the neck, followed by the trunk, has a moderately high load and, thus, a moderate ergonomic risk. Furthermore, the results of the ERP, the median values and the *relative time scores* show that the right and the left upper arm are least affected by incorrect strain and ergonomic risks.

It is, therefore, understandable that the evaluation of the median overall result shows that the posture of the tested students should be further investigated and possibly changed.

In the following, the relative time distribution of all RULA scores is shown in graphs for the different body regions.

The dental students spent 2% of their treatment time in a RULA score of 5, 19% in a RULA score of 6 and 79% in a RULA score of 7 (Figure 3).

When focusing on the different body regions, the posture of the trunk was found to be in a RULA score of 2 for 35% of the time, a RULA score of 3 for 31% and a RULA score of 4 for 34% of the treatment time (Figure 4). These score values suggest acceptable to moderate postures of the participants’ trunks when undertaking their tasks.

Figure 5 illustrates the percentage of treatment time spent in each RULA score and, thus, the posture for the neck region of the participants. The subjects’ neck was found to be in a RULA score of 3 for 51% of the time, with a RULA score of 4 for 40% and RULA score of 5 for almost 9% of the measured time. These score values suggest a moderate to poor posture for participants’ necks.

The left wrist was determined to be in RULA score 2 for < 1% of the measured time, in a RULA score of 3 for 5% of the time, RULA score of 4 for 33%, and, for 62% of the measured time, a RULA score of 5 (Figure 6). The right wrist was found to be in a RULA score of 2 for 1% of the measured time, a RULA score of 3 for 5%, a RULA score of 4 for 39% and a RULA score of 5 for 55% of the time. Based on these score values, both wrists had a very poor posture for most of the time, and the left wrist indicated a more unfavorable position for a longer time than the right wrist.

Figure 7 shows the relative time distributions of the lower arm’s left score and lower arm’s right score. The subjects’ left lower arm was in a RULA score of 1 for 1% of the time; for 33% of the time it was in a RULA score of 2 and for 66% of the time, it was in the worst possible posture (RULA score 3). This indicates that the left lower arm is in a poor posture for a long time and is ergonomically at a very high risk. The right lower arm of the participants was in a RULA score of 1 for 30% of the measured time, in a RULA score of 2 for 40% and in a RULA score of 3 for 30% of the time. These score values show that the left lower arm of the subjects adopted a more unfavorable position for a longer time than the right lower arm.

The left upper arm of the subjects was found to be in RULA 1 for 83% of the measured time, RULA 2 for 15% of the time, and, for 2% of the time, in RULA 3 (Figure 8). Since the maximum value for the upper arms extends to RULA 6, the left upper arm is in the good to moderate range in terms of the duration of poor posture. The right upper arm of the subjects was in RULA 1 for 32% of the measured time, RULA score 2 for 58% of the time, in RULA score 3 for 5% of the time and in RULA 5 < 1% of the time. Based on these scores, it can be seen that the right upper arm of the subjects held a less favorable position for a longer time than the left upper arm.

## 4. Discussion

The aim of this study was to assess the dental activities of students in terms of their ergonomic risk. Overall, the presented results show a high ergonomic risk, which can be attributed to the poor posture of the dental students. This can be seen from the values of the individual body parts measured in Table 4. The ergonomic risk classification results form the comparison of the ERP values. According to this, the left lower arm (ERP 0.84) has the highest risk, followed by the left and right wrist (ERP 0.78 and 0.68), which have a difference of only ERP 0.10. The left side is more affected than the right side. In contrast, the neck (ERP 0.53) and the trunk (ERP 0.46) have a lower ergonomic risk. The right and left upper arm (ERP 0.30 and 0.20) are the least affected by ergonomic risk.

We assume that the high score values and, thus, the high ergonomic risks for the left and right lower arm result from the rotation of the shoulder to reach the patient’s mouth. We also hypothesize that the reason for this is due to the compensatory movement of the lower arm during shoulder rotation. Therefore, the high ergonomic risk of the wrist is probably also due to the importance of this task during dental work. The right wrist performs precise repetitive movements in the mouth while holding weighty vibrating contra-angle handpieces, whereas the left wrist usually performs the static holding of the patient’s cheek and tongue; probably, these actions require the strong rotation of both wrists. We also assume that the poor visibility in the patient’s narrow mouth is responsible for the high ergonomic risk regarding the neck for dental students. To view the patient’s mouth, an unnatural rotation and protrusion of the neck is performed. The trunk, on the other hand, requires less rotation and flexion to perform the dental activity, which seems to minimize the ergonomic risk. It also stands to reason that the right and left upper arms are probably subjected to the least abduction and rotation and, thus, this also reduces the ergonomic risk.

Furthermore, our evaluations are confirmed by other studies [7,8] which found that dental students are at high risk of developing MSD due to their awkward posture. The RULA tool was also used, in part, for these studies [8,34], and also found a high to moderate risk of developing MSDs in the majority of dental students. Dable et al. [8] used the RULA method to evaluate the posture of students on different dental chairs and found that the values for the right side were higher and, therefore, the ergonomic risk was also higher in the right side. This contradicts our results, where we found a higher ergonomic risk for the left side (Table 4). This discrepancy can be explained by the absence of intraoral mirror use by the left hand of Dable’s subjects. As a result, they held the lower jaw of the dummy head with their left hand [8], whereas the subjects in our study use the mirror in their left hand, which causes the wrist to experience more rotation.

In addition, other researchers have found a high prevalence of MSD among dental students using the Nordic Questionnaire and emphasized the demand for ergonomic education [12].

Our measurement method provides an accurate and comprehensive approach in relation to the use of a kinematic analysis. The present results also provide information on the duration of the students’ harmful movement patterns, unlike other studies. Our results show that poor loading is exerted on the students’ bodies for almost 80% of the treatment time. The analysis of the inertial collected data allows for a detailed ergonomic risk assessment of all relevant body parts using RULA (Table 4).

The body regions affected by pain (MSD at the neck, shoulder and lower back) have been proven in many other studies [7,9,10,27,49]; however, they were not identified as the most affected by our analysis. Our postural analysis identifies the wrist and lower arm as having a higher ergonomic risk than the neck and trunk (Table 4) (Figure 4, Figure 5, Figure 6 and Figure 7). Therefore, the fact that carpal tunnel syndrome is recognized as a musculoskeletal occupational disease in German dentistry is an obvious reflection of this finding [50]. Likewise, questionnaire surveys given to dentists have shown that MSD also occurs in the hands and wrists [13,26,51]. The reason for the lower extremities and hands being at such a high ergonomic risk could be because the arm posture also affects the shoulder and neck region, as unnatural arm posture, in turn, leads to muscle overload [52]. Another explanation for the high ergonomic risk scores could be that RULA captures dynamic movements in addition to the statically held positions of the trunk; thus, the proximal upper extremities are also taken into account, as are the repetitive movements of the hands.

Other research showed that from 42.4% to 44.4% of dental students develop MSDs on the hand during their studies [11,12]. Here, too, the surveyed dentists mostly worked right-handedly and reported MSD more frequently for the right hand/wrist. While the study of Haas et al. [53] shows a high prevalence, especially for the right wrist, the present analysis specifies a worse posture for the left wrist. This discrepancy can be explained by the fact that other risk factors than just the body posture favor the occurrence of MSD in dentistry; these include performing precise movements with the hands or carrying out repetitive tasks over a long period of time [54]. Since nearly all subjects were right-handed [53,55], the right hand holds the heavy, vibrating instruments and performs very precise fine motor movements. This fact, in addition to the ergonomics, increases the risk for MSD and is, thus, responsible for the higher prevalence of pain in the right wrist [53,55]. The worse ergonomic risk for the left wrist/forearm, in contrast to the right wrist/forearm, can be explained by the predominantly static activity of the left side. This left-hand posture usually occurs simultaneously with a static procedure, e.g., when holding the cheek/tongue (the dummy heads did not possess a tongue) with the mirror. Due to these results regarding the wrist and the fact that pain-free, unrestricted wrist movements are necessary for accurate activity performance, special attention must be paid to this body region. Furthermore, the fact that the subjects are still students who are only at the very beginnings of their career should not be neglected. In this regard, the question of prophylactic and pain-relieving therapies should be investigated in further analyses.

The present results of such high ergonomic risk levels raise the question of the reasons for these findings. We assume that the subjects have little knowledge about ergonomics and its practical implementation, since there is a lack in the provision of ergonomics training in dental studies. This fact of poor postural awareness among dental students has previously been widely reported, including in studies using surveys [10,31,35,36,37]. The poor posture may also be traced to the untrained muscles of the subjects and, similarly, external factors (the patient’s chair, dentist’s chair, equipment and cabinets) may also have an influence on ergonomics [44].

The study limitations are as follows: since the measurements were performed in a standardized manner under laboratory conditions and adapted to realistic conditions, recording a more optimal routine workflow from an actual dental practice is not possible, as the students do not practice in their usual environment. Furthermore, measurements were not taken during work on real patients, but on a dummy head, to ensure the standardization of the work processes and achieve a better comparison. Further research is needed to obtain specific comparisons between the external factors and their degrees of influence on the ergonomic posture of the students.

Regarding ergonomics in dentistry, it is important to know whether the different quadrants have an influence on the ergonomic risk, and which dental activities have a higher risk. For our measurements, each quadrant was assigned a specific task; therefore, the influence of the quadrants on the students’ ergonomic risk cannot be precisely analysed in this study. If every task had been performed in each of the four quadrants, it would have exceeded the timeframe. By having subjects perform a different task per quadrant, we provided a range of versatile tasks that correspond to real, everyday practice. At the same time, the factors of physical exertion and the concentration of the subjects were not negatively affected due to the duration of execution; therefore, this methodology allows a range of dental methods to be better represented Further kinematic analyses should, nevertheless, investigate the ergonomic differences between the quadrants and dental activities to gain further insight into the reasons for poor posture in dentistry.

In addition, an important question to address is whether the age or level of job experience correlate with ergonomics in dentistry. Through our analysis, it was determined that high ergonomic risk levels are not established after many years of workload, but that high ergonomic risk potentials arise due to possible unfavorable movement patterns, which occur as early as when first studying dentistry. In order to allow for more accurate comparisons between age and job experience levels, a further analysis of experienced, older dentists should be carried out.

Furthermore, in order to prevent faulty posture and the resulting postural defects, such as MSD, targeted training of the crucial muscle groups would be beneficial. Regular stretching, strength training and physical therapy can reduce the prevalence of MSD [56,57]. To determine the improvements gained from such training, further studies should measure the posture before and after muscle training.

By using the newest technology, we have been able to provide tangible evidence for the high ergonomic risk posed to students during dental activities. These alarming results should demonstrate the importance of teaching ergonomics in the curriculum to universities. At German universities, ergonomics education is often a single unit, integrated into a degree program, which is not thorough enough and, therefore, obviously ineffective. Therefore, in order to implement an effective education, the theory and practice of ergonomic posture should be more extensive and be included within occupational health training courses for the benefit of the students. Commencing in college and continuing up to retirement, the dental profession requires good physical and mental performance. Consequently, it is important to teach MSD prevention in dental school in order to reduce the risks involved, as well as the high prevalence and persistent symptoms of MSD. Only in this way can dental students and, in the long term, dentists, achieve good physical and mental vitality.

## 5. Conclusions

The kinematic analysis of dental activities shows that dental students are exposed to a significantly high ergonomic risk during their dental work. In particular, the wrists of dental students exhibited a particularly high ergonomic risk in contrast to other body parts. We were able to demonstrate that the left side of the body is more affected by ergonomic risk than the right side of the body. Thus, we confirm existing results [7,8,9,10,11], which were mainly collected via questionnaires, with our ergonomic risk assessment based on kinematic data. These existing studies [7,8,9,10,11] found that musculoskeletal disorders (MSDs) are already widespread among dental students, despite the short duration of their dental practice. These questionnaire analyses showed that the upper body, with the neck and upper extremities, is particularly at risk [7,9,11,13]. Since high ergonomic risk levels are often correlated with MSD, this indicates a high risk for developing MSD. It is necessary for students to be educated about the ergonomic risks at work and learn how to prevent them. Universities should give high priority to the importance of preventive measures and ergonomics teaching [2,31,58]. In addition, future analyses should be conducted to optimize dental equipment and targeted preventive muscle training.

## Figures and Tables

**Figure 1 ijerph-18-10550-f001:**
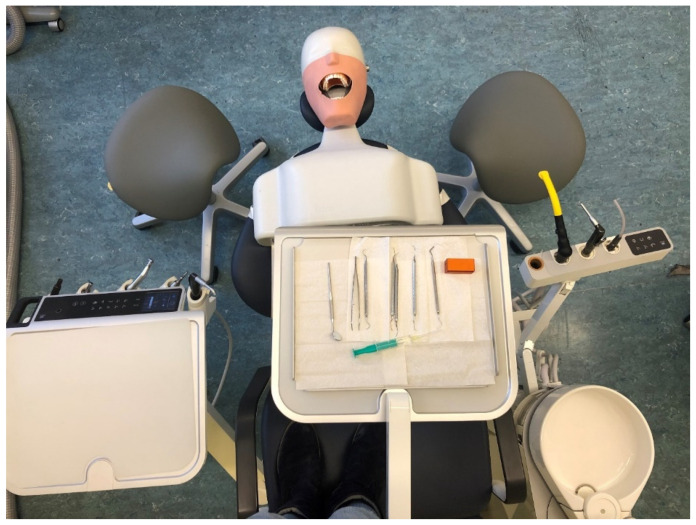
Structure and adjustment of dental treatment concept one. During each dental activity, the dental students used magnifying glasses.

**Figure 2 ijerph-18-10550-f002:**
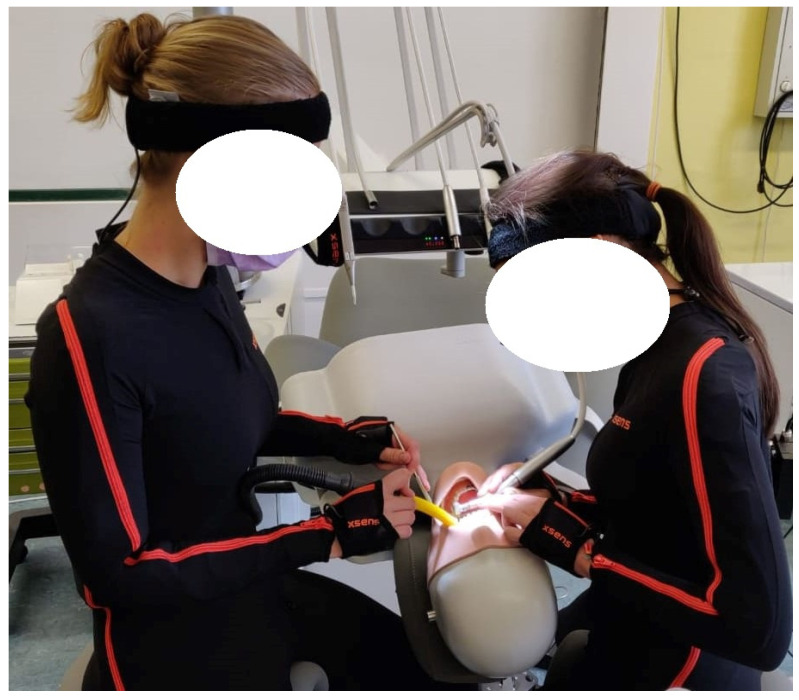
Exemplary treatment on the dummy head of 2 study participants. Both wear the measuring suit with the integrated sensors.

**Figure 3 ijerph-18-10550-f003:**
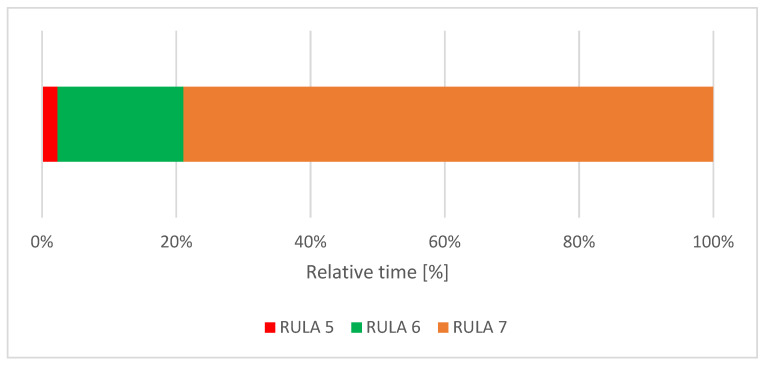
Relative time distribution of the total RULA score.

**Figure 4 ijerph-18-10550-f004:**
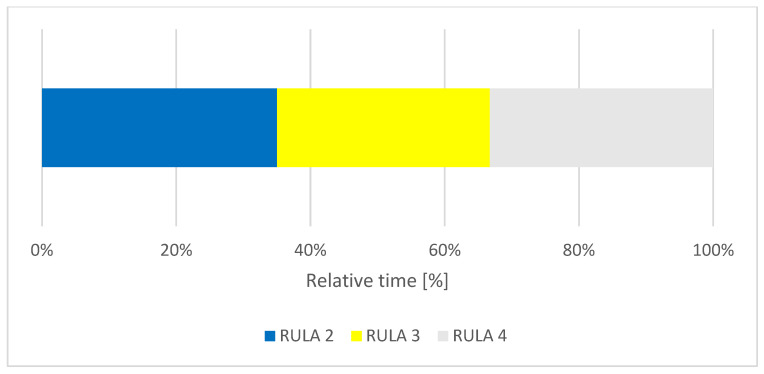
Relative time distribution of the trunk score.

**Figure 5 ijerph-18-10550-f005:**
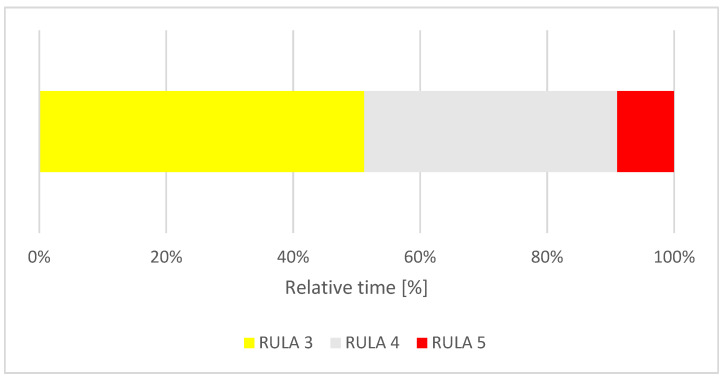
Relative time distribution of the neck score.

**Figure 6 ijerph-18-10550-f006:**
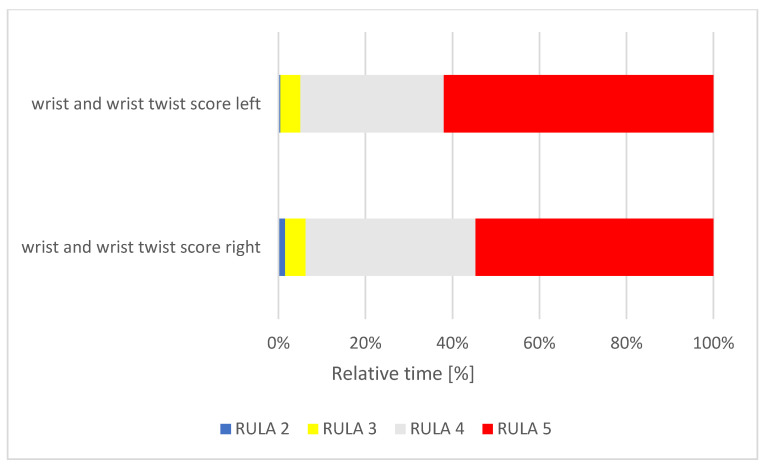
Relative time distribution of the wrist scores.

**Figure 7 ijerph-18-10550-f007:**
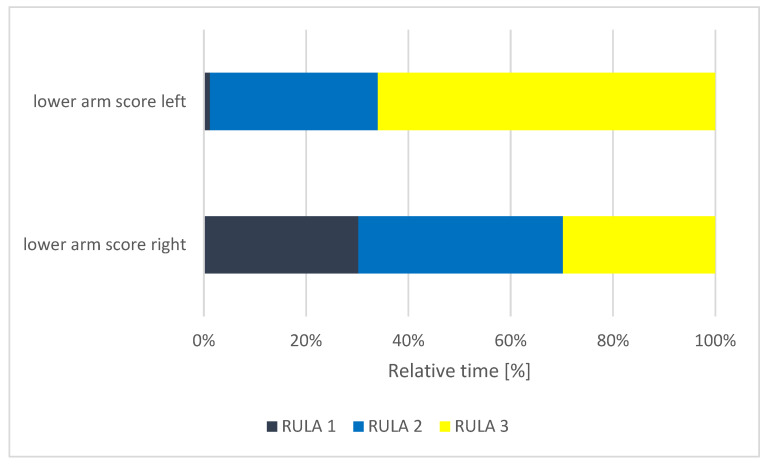
Relative time distribution of the lower arm scores.

**Figure 8 ijerph-18-10550-f008:**
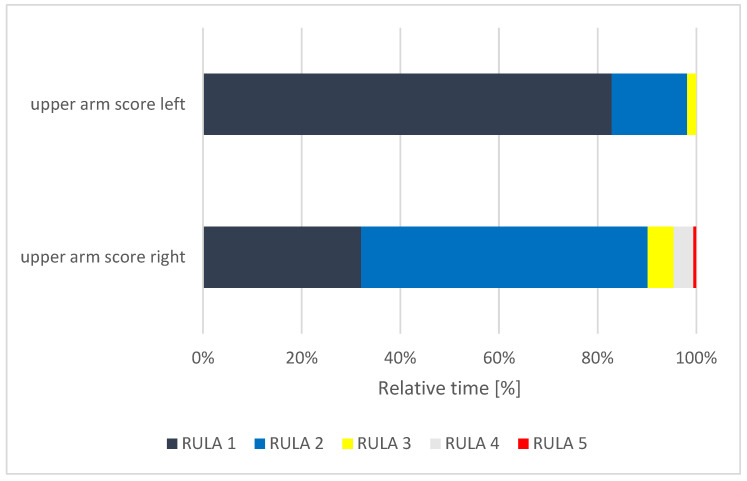
Relative time distribution of the upper arm scores.

**Table 1 ijerph-18-10550-t001:** Demographic characteristics of all participants. In addition to the number of participants (n), the interquartile range (IQR) is also shown as needed. * DSs: Dentistry students, * DAs: Dental assistant trainees.

Personal Characteristics	DSs *	DAs *
Sex		
Female	12 (80%)	13 (86.7%)
Male	3 (20%)	2 (13.3%)
Median age (years)	27 (2.5)	21.5 (4)
Median height (cm)	170 (18)	165.5 (6.25)
Median body weight (kg)	68 (14.5)	59 (8)
Handedness		
Right	15 (100%)	15 (100%)
Left	0 (0%)	0 (0%)

**Table 2 ijerph-18-10550-t002:** Standardized dental tasks performed by the subjects.

	Task	Quadrant 1	Quadrant 2	Quadrant 3	Quadrant 4
General Dentistry/Students		Tooth filling of tooth 16	Preparation of tooth 26 for crown uptake	Root canal treatment on tooth 35	Tartar removal in the 4th quadrant
	1	Prepare tooth cavity with a cylindrical diamond bur and the use of wedges	Occlusal reduction using an occlusal reducer	Perform an entrance cavity and trepanation on tooth 35 using a diamond-coated cylinder	Removal of supra- and subgingival tartar/calculus using scalers an curettes
	2	Create a Tofflemire die using a die clamp	Chamfer preparation using a torpedo-shaped diamond burand approximal reducer	Find the channel entrance using an endo file	
	3	Tooth filling with ketac^®^ while using a ketac^®^-set and a cougar/heidemann		Manual preparation of the canal using an ISO 20–40 endo file with regular irrigation using a irrigation cannula	

**Table 3 ijerph-18-10550-t003:** RULA modifications. We refer to the steps defined by McAtamney and Corlett [45].

Worksheet-Steps	Parameters	Modifications of the RULA Parameters	Additional Information
STEP 1	Leaning arm	Since the arms of the dental students were not supported at any time, 0 was given as the score of the leaning arm.	
STEP 3	Wrist is bent laterally	If the lateral bend of the wrist was less than −10° (radial deviation) or more than 10° (ulnar deviation), +1 was added to the wrist score.	The RULA score does not show how much the wrists need to be bent.
STEP 4	Wrist twist	If the wrist twist was in the neutral range (45° to −45°), +1 was added to the wrist twist score. If the turn of the wrist was close to the terminal range of motion (90° to 45° and −45° to −90°), +2 was added to the wrist twist score.	The RULA score does not show how much the wrists need to be twisted.
STEP 6	Muscle use score of arm and wrist	The score was increased by +1 for static or repetitive muscle work. - Muscle work was classified as static if the difference in angular velocity of the shoulder joint at the beginning and end was ≥7.5 for longer than 10 s [48]. - Muscle work was classified as repetitive if the movement of the joint indicated more than 0.5 Hz mean power frequency [48]. For this purpose, extension and flexion of the wrist, as well as forearm rotation, were taken into account.	The given scores from the RULA table were transformed into continuous recordings.
STEP 9 + 10	Twist trunk and neck	If a rotation of the neck or trunk was deduced to be inferior to −10° or superior to 10°, a score of +1 was added to the ‘neck score’ or ‘trunk score’.	In order to determine the exact extent of the movement, certain scores were added.
STEP 9 + 10	Side bending trunk and neck	If the neck or torso inclination (in the frontal plane) was more than +10° or less than −10°, +1 was added to the neck or torso score.	In order to determine the exact extent of the movement, certain scores were added.
STEP 11	Supported legs and feet	As the legs and feet of the dental students were permanently supported by their sedentary work, +1 was given as the supported leg score.	
STEP 13	Muscle use score of neck, trunk and legs	The score was increased by +1 for static or repetitive muscle work. Muscle work was classified as static if the difference in angular velocity of the neck/cervical spine and lower back/lumbar spine was ≥7.5 at the beginning and at the end for longer than 10 s. - Muscle work was classified as repetitive if the movement in any of the degrees of freedom of the joint indicated above 0.5 Hz mean power frequency.	
STEP 14	Force/load score	Since all dental instruments weighed less than 2 kg, this score was set to 0 [48].	

**Table 4 ijerph-18-10550-t004:** Median, relative share of the *relative time score* and ERP for all evaluated RULA steps.

RULA Score	Median (IQR)/Max. Score	Relative Time Score	ERP
Final Score	7 (0)/7	6.67	0.95
Trunk Score	3 (2)/6	2.74	0.46
Neck Score	4 (1)/6	3.15	0.53
Right Wrist Score	4 (1)/6	4.06	0.68
Right Lower Arm Score	2 (2)/3	1.96	0.65
Right Upper Arm Score	2 (1)/6	1.82	0.30
Left Wrist Score	5 (1)/6	4.66	0.78
Left Lower Arm Score	3 (1)/3	2.53	0.84
Left Upper Arm Score	1 (0)/6	1.21	0.20

## Data Availability

The data presented in this study are available on request from the corresponding author.

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
