# Peer review of "Ergonomic Risk Assessment of Dental Students—RULA Applied to Objective Kinematic Data"

_ijerph, 2021, doi:10.3390/ijerph181910550_

Round 1

Reviewer 1 Report

  1. Table 1, please address a note about the DSs and Das.
  2. The format of the paper is very messy and leads to difficulty of reading, especially in the caption of figures and tables.
  3. It would be good if the authors provided a picture of the participant with 17 sensors when doing a dental treatment.
  4. Ln 32. The authors used 20 references here. I doubt the quality and correction of the 20 references. Please modify.
  5. There are many MSDs assessment methods (REBA, KIM, OWAS…). It should be mentioned in the Introduction. Moreover, to explain why the authors eventually apply LULA in the study?
  6. Many sentences and statements in the Introduction section are repetitive and redundant. Please revise.
  7. Ln 174-175. Do the authors mean that the movement of the treatment chair is not allowed? The setting is not similar to an actual situation in dental treatment.
  8. The main finding of the study is “ The kinematic analysis of dental activities shows that dental students are exposed to high ergonomic risk during their dental work. In particular, the wrists of dental students showed a high ergonomic risk.” Much literature had reported the same results, as the authors mentioned in the Introduction. What is the new information and contribution of the current study?
  9. Please consider using MSDs to represent musculoskeletal disorders.

Author Response

  1. Table 1, please address a note about the DSs and Das.

-->Done

  1. The format of the paper is very messy and leads to difficulty of reading, especially in the caption of figures and tables.

--> Thank you for pointing this out. Now that we have received a formatted version from the publisher for the corrections, this has hopefully been resolved.

  1. It would be good if the authors provided a picture of the participant with 17 sensors when doing a dental treatment.

--> We integrated Figure 2 showing two participants wearing the suit while doing a dental treatment.

  1. Ln 32. The authors used 20 references here. I doubt the quality and correction of the 20 references. Please modify.

--> We modified this sentence and deleted references. Since our Endnote program can only make these changes to references without the tracking mode, this is unfortunately not immediately visible here.

  1. There are many MSDs assessment methods (REBA, KIM, OWAS…). It should be mentioned in the Introduction. Moreover, to explain why the authors eventually apply LULA in the study?

-->Done

  1. Many sentences and statements in the Introduction section are repetitive and redundant. Please revise.

--> We have completely revised the introduction and tried to delete everything redundant.

Ln 174-175. Do the authors mean that the movement of the treatment chair is not allowed?

 --> The subjects were asked to adjust the chair adequately before the measurements. During the measurements, a change in the seat setting was not permitted. This does not mean that no movements on the chair were allowed during the measurements.

  1. The main finding of the study is “ The kinematic analysis of dental activities shows that dental students are exposed to high ergonomic risk during their dental work. In particular, the wrists of dental students showed a high ergonomic risk.” Much literature had reported the same results, as the authors mentioned in the Introduction. What is the new information and contribution of the current study?

-->Done

  1. Please consider using MSDs to represent musculoskeletal disorders.

-->We have now used the abbreviation MSD throughout the document.

Reviewer 2 Report

The article entitled "Ergonomic risk assessment of dental students – RULA applied to objective kinematic data" presents the analysis of dental students' posture while performing dental procedures and its inclination for further musculoskeletal disorders. The authors emphasise the high ergonomic risk of dental students during their dental work and the need of appling preventive measures and ergonomics teaching during the education.

The article needs only minor corrections:

- line 24  - please change Rula into RULA

- line 41 – please remove double space

- line 75 – please remove double space

- line 90 - please remove double space

- line 106 - please remove double space

- line 112 – “Therefore, in the present study, a combination of ergonomic risk assessment (RULA)” should there be “Rapid Upper Limb Assessment” instead  of ergonomic risk assessment

Table 1 – please elaborate the abbreviations DSs and Das

- line 155 - please remove double space

- line 157 – “the subjects were sitting on a dental chair” was it a dental chair or dentist chair.

Line 162 – “After calibration, the recordings of the dental activities made on a dummy head” please rephrase this sentence.

Line 195 - please remove double space

Line 205 – “may need changes” please start the sentence with the capital letter

Where all the participants right-handed? This information should be added to the “Material and methods” section

271 - please remove double space

Line 283 “Table 2. for < 1% of the time measured, in RULA score 3 for 5% of the time, in RULA 283 score 4 for 33% and for 62% of the measured time in RULA score 5 (Figure 5).” – please rephrase this sentence. Looks like “table 2” should not be here

Line 359 - please remove double space

Author Response

The article needs only minor corrections:

- line 24  - please change Rula into RULA.

-->Done

- line 41 – please remove double space

-->Done

- line 75 – please remove double space

-->Done

- line 90 - please remove double space

-->Done

- line 106 - please remove double space

-->Done

- line 112 – “Therefore, in the present study, a combination of ergonomic risk assessment (RULA)” should there be “Rapid Upper Limb Assessment” instead  of ergonomic risk assessment.

--> We changed this sentence as follows: Therefore, in the present study, a combination of Rapid Upper Limb Assessment as one ergonomic risk assessment and inertial motion capture technology…

Table 1 – please elaborate the abbreviations DSs and Das.

-->Done

- line 155 - please remove double space.

-->Done

- line 157 – “the subjects were sitting on a dental chair” was it a dental chair or dentist chair.

-->Thank you for pointing this mistake out. We mean “dentist chair”.

Line 162 – “After calibration, the recordings of the dental activities made on a dummy head” please rephrase this sentence.

-->We corrected the sentence.

Line 195 - please remove double space.

-->Done

Line 205 – “may need changes” please start the sentence with the capital letter.

-->Done

Where all the participants right-handed? This information should be added to the “Material and methods” section.

-->Yes, it is shown in table 1 and we added this information in the description of the subjects.

271 - please remove double space.

-->Done

Line 283 “Table 2. for < 1% of the time measured, in RULA score 3 for 5% of the time, in RULA 283 score 4 for 33% and for 62% of the measured time in RULA score 5 (Figure 5).” – please rephrase this sentence. Looks like “table 2” should not be here.

àThank you for pointing our mistake out. We changed this sentence as follows: The left wrist was determined to be in RULA score 2  for < 1% of the time measured, in RULA score 3 for 5% of the time, in RULA score 4 for 33% and for 62% of the measured time in RULA score 5 (Figure 5).

Line 359 - please remove double space.

-->Done

Author Response

#1: Read and correct each sentence in order to improve the logic and clarity of the text.

-->Following a comment from another reviewer, we have revised the introduction and deleted redundant statements.

E.g., line 15: You stated that MSDs are already common among dental students. What do you mean by “already”?

-->We mean the young age and short period of dental activities so far during a dental study. We corrected the sentence. We corrected this sentence as follows: Musculoskeletal disorders (MSDs) are already prevalent in dental students despite their young age and short duration of dental practice.

#2: Read and correct each sentence in order to improve the text linguistically. E.g., line 16: You state that “Current findings stated…”; the verb must be in the present tense.

-->Done

3: Eliminate unnecessary spaces through the text. E.g., line 186 “view ,”and line 195 “score included”.

-->Done

#4: Specify, from the abstract, the handedness of the subjects.

-->Done

 Also consider expressing and analysing results using “dominant” and “non-dominant” wrist/upper arm/lower arm/etc., instead of right and left wrist/upper arm/lower arm/etc.

-->All subjects were right-handed. Therefore, we assume that the right side of the body is always the dominant side. However, this does not imply that all of them really always treated according to the same scheme. Because the way of performing each activity was up to each participant. This means in the reverse conclusion also that not everyone necessarily treats as right-handed also always equally. Since we cannot assume with certainty that the right side is always the dominant side, we would like to avoid this designation and keep right and left as designation.

#5 (line 23): “All relevant body parts combined” is not a clear expression.

-->Done

Also, consider replacing “had the worst possible posture that could be achieved” with “had a posture with the highest RULA score”.

-->Done

#6 (line 24): Replace “Rula” with “RULA”.

-->Done

#7 (line 26): Is training the only way to reduce risk?

-->We meant training ergonomics. But since this seems to lead to misunderstandings, we have deleted this word.

#8 (line 115): [59] is cited incorrectly because it does not pertain to the previous sentence. You could modify it by specifying for example “… using a phantom head, replicating a similar study conducted on dentists by Maurer-Grubinger et al. [59].

-->Thank you for pointing this out. We changed it.

#9 (line 122): “Fifteen teams each consisting of one dental student and one dental assistant trainee” might be clearer than “Fifteen teams of dental students and dental assistant trainees”.

-->done

#10 (table 1): Define acronyms (DSs and Das) in the caption.

-->Done

#11 (section 2.2): Have you combined motion capture system with videotaping?

-->We additionally filmed the entire measurements. This was only for backup purposes to better identify and explain kinematic "outliers" in case of emergency. The video recordings were not synchronized with the kinematic measurements. We have therefore formulated this sentence in a misleading way and have now corrected it.

#12 (sections 2.2 and 2.4): Were the RULA scores calculated every twenty-fourth of a second?

-->No. There were 24 recordings stored every second. At the end of each activity, the mean value was calculated from the sum of each measurement recording.

#13 (section 2.3): Clarify the mean of concept, quadrant, and task

-->We clarified it in the table description of table 2.

(see line 122: “… four typical tasks”). Identify clearly the tasks performed by students and assessed by RULA.

-->We have deleted this text insertion because the explanation follows later at table 2.

In addition, describe the activities performed by the right (dominant) hand and the left (non-dominant) hand during each task specifying both working tools and other equipment (e.g., magnifying glasses) used.

-->Unfortunately, this comment cannot be implemented in our opinion, since we have left it up to the test persons to decide how they perform the work (see #4). This simulates the practical situation. Only they were forbidden to move the dummy head during the measurements. The dummy head was only allowed to be adjusted beforehand. In the content description of Table 2 we have added the approximate duration of the activities.

#14 (line 164): Consider replacing “Since dental treatment concept 1 is exclusively taught …” with “Since only dental treatment concept 1 is taught …”.

-->Done

#15 (table 2): [60] is cited incorrectly. You could modify it by specifying for example “… concept 1 defined by Ohlendorf et al. [60]”.

-->Done

#16 (figure 1): Move the caption under the figure.

-->Done

Addition, move the sentences “The setting of the lamp… to be changed.” in the text of the paper.

-->Done

#17 (line 175): Remove [60].

-->We do not find this reference in this line. If the method paper of Ohlendorf et al. should be meant, this reference must remain here, since a more detailed description about this topic can be read here.

#18 (section 2.4): Consider removing “evaluation” after “RULA” since RULA means Rapid Upper Limb Assessment; in other words, avoid “assessment evaluation”.

-->Done

#19 (section 2.4): Motivate the choice of RULA as evaluation method.

-->Done

#20 (lines 186-209): Refer only to the original study [62].

-->We corrected it.

#21 (references): Check all references. E.g., [62] and [64] are the same paper.

-->Thank you for pointing this out. We checked it.

#22 (tables): Check the numbering of the tables.

-->Thank you for this hint.

#23 (table 4): Improve the table in terms of punctuation and spacing. Move the sentences “This table… approach [65,66].” from the caption to the text of the paper.

-->Done

Explain in the caption that you refer to 2 the steps defined by McAtamney and Corlett [62].
-->Done

Check all sentences in the table. E.g., “If the lateral bend of the wrist was less than 10°…” should be “If the lateral bend of the wrist was less than -10°…”; “0 was given as the supported leg score” should be “+1 was given as the supported leg score”.
-->Thank you for pointing this mistake out.

#24 (lines 212-231): Detail the meaning and the method of calculating the outcomes. In particular: 1) specify whether they were determined for the set of tasks carried out by all students (evaluating individual tasks could be interesting)

-->In this study, a different task was performed in each quadrant. Therefore, the comparison of the quadrants is not possible here. But we have already made a note of this further analysis approach for follow-up studies.

and 2) since the “original” RULA allows the worst posture at one point in time to be assessed, justify or demonstrate the validity of the relative time score, of ERP and of maintaining the “original” risk classification. Regarding 2), are you sure that, for example, 95%×1+5%×7=1.3 is acceptable and better than 25%×1+75%×2=1.75?

Relative time score, ERP, and the original risk classification are correlated, with:

ERP is directly calculated from the relative time score:  rho =1  , p<0.0001,

Relative time score, original risk classification:  rho = 0.59, p=0.0232

ERP, original risk classification: rho = 0.59, p=0.0232

This underlines the validity of the relative time score and the ERP compared to the original risk score.

You are correct in your point and we will consider this issue in further analysis, although your example is less realistic.

#25 (line 213): Add “, relative to both total RULA score and individual body region RULA scores” after “… based on three outcomes”.

-->Done  

Use only two phrases (e.g., “total RULA score” and “individual body region RULA scores”) throughout the text.

-->We have corrected it throughout the text

#26 (line 234): Specify the meaning of IMC.                                    

-->Sure we mean IMU. Thank you for pointing this mistake out.

#27 (sections 3.1 and 3.2): Remove the section 3.2 and present the data in a table having, for example: scores (1, 2, …, 7) in columns; body regions (total, trunk, neck, …) in rows; percentage time spent in cells. Remove the title of section 3.1 (in other word, keep only section 3. Results).

-->We have deleted the headings 3.1. and 3.2. These data are already in the current table 4, but we want to keep the order of the tables and graphs. Especially the graphs illustrate the data better than leaving only table 4.

#28 (section 3): Insert a time trend of the total RULA score or a body region RULA score of the tasks performed by a student, as an example. In this section, you could present, in order: the figure of this time trend, the new table (see #27), the current table 5. Also specify the durations of the tasks performed by students.

-->We have added the approx. duration of each task to the description of table 2. Unfortunately, we do not see the added value if we only take one person as an example. This way, we have averaged the data over all of them and generate much more significance in our representations.

#29 (section 3): Avoid repeated or contradictory comments. E.g., lines 252-256.

-->Please be more specific. In our opinion some phrases may sound the same, but they are different in their message, because different parameters are quoted.

#30 (lines 241-242): You state that “The maximum number of points… shows how high the ergonomic risk is in each case”. Are you sure?

-->I have corrected it and above all analysed ERP to analyse the ergonomic risk

#31 (line 246): Check the use of “consequently”.

-->Done

#32 (lines 254-256): You state that the relative time scores reveal that the left lower arm has the highest ergonomic risk, but this is not consistent with Table 5.

-->Done

#32 (section 3): Why do you not analyse and comment, only or mainly, on ERP?

-->Thank you for pointing this out. We have tried to implement this suggestion.

#33 (section 3): Give more detailed reasons why the left limb is more at risk than the right one, also analysing the RULA parameters/steps.

-->In the results section, we only present our results and do not interpret why the left limb is at higher risk. We have discussed this topic in the discussion.

#34 (lines 312-318): Specify the mean of ergonomic risk (i.e., median RULA score).

-->We have explained this at the appropriate place. What is meant by this is that 24 recordings are captured every second. The mean value is then calculated from the sum of all recordings at the end of each task.

#35 (lines 318 and 336): Replace “upper body” with “trunk”.

-->Done

#36 (line 321): Replace “possibly” with “possible”.

-->Done

#37 (lines 321-340): Why do you not focus on ERP instead of the median RULA score

-->We rephrased the results and discussion section. Now, we focus on both.

#38 (line 324): Are you sure that 2/3 is worse than 5/6, and that 2/3 is worse than 4/6? Why do you prioritise the absolute differences between the median values and the maximum values instead of the relative differences? However, these are results (section 3).

-->Thank you very much for this note. For such statements we have calculated the ERP and therefore this is now also improved in the discussion.

We corrected as follows: The ergonomic risk classification results from the comparison of the ERP values. According to this, the left forearm with the highest ERP value (ERP 0.84) has the highest risk, followed by the left wrist (ERP 0.78) and the right wrist (ERP 0.68), which each have a difference of only ERP 0.10 to each other. The right forearm (ERP 0.65) follows with only ERP 0.03 difference. This is followed by the neck (ERP 0.53) and upper body (ERP 0.46) with a lower ergonomic risk. While lastly follows the right upper arm (ERP 0.30) and left upper arm (ERP 0.20) with the lowest ergonomic risk.

#39 (lines 327-330): You state that “the high ergonomic risk for left and right forearm result from the rotation of the shoulder”. Does the shoulder position affect the forearm position? Explain.

--> We added the following explanation: We also hypothesize that the reason for this is due to the compensatory movement of the forearm during shoulder rotation.

#40 (line 331): what do you mean by “heavy”?

-->weighty…We corrected the word

#41 (lines 332 and 338): Check the use of “presumably” and “likely”.

-->We corrected the word

#42 (lines 336-338): Check the sentence.

-->Done

#43 (line343): Replace “RULA risk assessment (RULA tool)” with “RULA tool”.

-->Done

Reviewer 4 Report

The work is interesting, needs some minor editorial porations. I recommend that you read the instructions for authors (https://www.mdpi.com/journal/ijerph/instructions). There is a significant number of self quotations that need to be removed and a lack of sample size calculations or calculate the power of the test.

  1. The work contains numerous self-citations:

  • Fabian Holzgreve - #59/60/69/73
  • Laura Fraeulin - #59/69/73
  • Christina Erbe - #45/59/60/69/73/77
  • Werner Betz - #59/60/69/73
  • Eileen M. Wanke - #59/60/69/73
  • Doerthe Brueggmann - #59/69/73
  • Albert Nienhaus - #20/39/59/60/73
  • Christian Maurer-Grubinger - #59/60
  • David A. Groneberg - #45/77
  • Daniela Ohlendorf - #45/59/60/69/77

Reduce the number of self quotations.

  1. affiliations - add unit postcodes.
  2. L31-32 - You cite 20 items to support your opinion. This is a mistake. Cite up to a maximum of 5 studies. My comment applies throughout the text.
  3. Table 1. - 5- it does not conform to the mpdi J. Environ. Res. Public Health style, correct it according to the guidelines.
  4. https://www.mdpi.com/journal/ijerph/instructions
  5. L151 - "Xsens full-body suit" - suggests adding a photo showing the suit.
  6. Figure 1 - 7 - the signature should be under the figure.
  7. Statistical Analysis - Please add sample size calculations or calculate the power of the test.
  8. L283 - why "tables. 2" is in bold?
  9. L385-412 - The entire paragraph refers to the authors' research. Suggests removing it adding research not related to the authors. This increases the scientific integrity of the paper.
  10. Add clear limitations to the study at the end of the discussion.
  11. L475 - suggests deletion - you develop abbreviations in the text.
  12. Author Contributions - The following statements should be used "Conceptualization, X.X. and Y.Y.; Methodology, X.X.; Software, X.X.; Validation, X.X., Y.Y. and Z.Z.; Formal Analysis, X.X.; Investigation, X.X.; Resources, X.X.; Data Curation, X.X.; Writing – Original Draft Preparation, X.X.; Writing – Review & Editing, X.X.; Visualization, X.X.; Supervision, X.X.; Project Administration, X.X.; Funding Acquisition, Y.Y.”,

https://www.mdpi.com/journal/ijerph/instructions

Author Response

The work is interesting, needs some minor editorial porations. I recommend that you read the instructions for authors (https://www.mdpi.com/journal/ijerph/instructions). There is a significant number of self quotations that need to be removed and a lack of sample size calculations or calculate the power of the test.

-->We are aware that self-citations in scientific papers should rather be avoided. However, since this group of authors is the only one that has done this kind of research so far, it is necessary to refer to it. Thus, we have referred to 2 method papers, in which on the one hand the method and on the other hand the evaluation procedure are described in more detail. Another article on current MSD prevalences of the hand is also necessary to refer to, as no other article has published such new data and differentiates between left and right hand.

Two other sources, where A. Nienhaus is a co-author, were published by another research group/institute. These reviews are also the most recent available nationally. All other sources have been deleted.

  1. The work contains numerous self-citations:

  • Fabian Holzgreve - #59/60/69/73
  • Laura Fraeulin - #59/69/73
  • Christina Erbe - #45/59/60/69/73/77
  • Werner Betz - #59/60/69/73
  • Eileen M. Wanke - #59/60/69/73
  • Doerthe Brueggmann - #59/69/73
  • Albert Nienhaus - #20/39/59/60/73
  • Christian Maurer-Grubinger - #59/60
  • David A. Groneberg - #45/77
  • Daniela Ohlendorf - #45/59/60/69/77

Reduce the number of self quotations.

  1. affiliations - add unit postcodes.

-->done

  1. L31-32 - You cite 20 items to support your opinion. This is a mistake. Cite up to a maximum of 5 studies. My comment applies throughout the text.

-->Done

  1. Table 1. - 5- it does not conform to the mpdi J. Environ. Res. Public Health style, correct it according to the guidelines.

-->We have received an already pre-formatted format back from the publisher, in which the table format has also been adjusted.

  1. L151 - "Xsens full-body suit" - suggests adding a photo showing the suit.

--> We have added Fig. 2

  1. Figure 1 - 7 - the signature should be under the figure.

-->Done

  1. Statistical Analysis - Please add sample size calculations or calculate the power of the test.

-->In consultation with our statisticians, we did not perform a power calculation because this is a pilot study of a descriptive nature. As we did not conduct statistical tests in this descriptive study, we only estimated the expected precision of the estimates in advance: For n=20 the length of the 95%-confidence interval for the mean would be 0.95 of the estimated standard deviation (sd), hence for sd=1 the length of the 95%-CI is 0.95. We have had a total of these 40 subjects (20 teams) measured. But due to incomplete data sets of one team member, we had to exclude 5 teams from the final analysis in the end. We integrated this information into the methods section.

  1. L283 - why "tables. 2" is in bold?

-->It was a mistake, we corrected the sentence.

  1. L385-412 - The entire paragraph refers to the authors' research. Suggests removing it adding research not related to the authors. This increases the scientific integrity of the paper.

-->We rephrased this paragraph.

  1. Add clear limitations to the study at the end of the discussion.

--> We have already mentioned limitaitons at the end of the discussion. Since they seem not to have been identified as such, we have now made it clear by means of an introduction.

  1. L475 - suggests deletion - you develop abbreviations in the text.

-->done

  1. Author Contributions - The following statements should be used "Conceptualization, X.X. and Y.Y.; Methodology, X.X.; Software, X.X.; Validation, X.X., Y.Y. and Z.Z.; Formal Analysis, X.X.; Investigation, X.X.; Resources, X.X.; Data Curation, X.X.; Writing – Original Draft Preparation, X.X.; Writing – Review & Editing, X.X.; Visualization, X.X.; Supervision, X.X.; Project Administration, X.X.; Funding Acquisition, Y.Y.”,

https://www.mdpi.com/journal/ijerph/instructions

--> We changed it.

#44 (section 4): Synthesise comparison with literature. In particular: 1) select only studies on dental students; and 2) connect all studies with similar results, avoiding repetitions.

-->Thank you for pointing this out. We rephrased the introduction and discussion.

#45 (lines 397-401): This sentence seems to contrast with others.

-->Since nearly all subjects were right-handed, the right hand, thus, holds the heavy, vibrating instruments and performs very precise fine motor movements[69,73].

#46 (lines 422-423): Clarify the sentence “it may not possible…”.

-->We rephrased this sentence.

#47 (lines 430-439): Clarify the sentences between “If every… of poor posture in dentistry.”.

--> We changed “risks” to “reasons”.

48 (line 457): [77] seems an incorrect citation. Moreover, you state that “the ergonomics education… is, evidently, ineffective”. Are you sure that the problem/solution is only the training? (See also #7).

-->We rephrased this sentence.

#49 (Abbreviations): Is this section necessary? If yes, add all abbreviations (e.g., ERP and RULA), and list them in alphabetical order. Moreover, replace “MSD - musculoskeletal disorders” with “MSD - musculoskeletal disorder”, and “SOpEZ with SOPEZ”.

-->It´s not necessary, we replaced all terms.

Round 2

Reviewer 1 Report

Thank you very much for your polite responses and considerable edits to amend this manuscript. I believe it is now acceptable for publication, following just a few very minor edits.

1. In the current form, many typos needs to correct. The authors should correct them in the proofreading stage.

Author Response

We proofread the text already and corrected the typos.

Reviewer 3 Report

The paper has improved. However, I have some comments.

First of all, it is necessary to re-read the text. It is embarrassing for me to suggest deleting the sentence "Klicken oder tippen Sie hier, um Text einzugeben" (lines 60 and 452).

Re-read the whole text, also to correct punctuation. E.g., line 370 “analysisThe present”, line 389 “[13,14].Here”, line 337 “0.68) , which”, line 213 “[50.]”.

Referring to my previous comments:

#12 + #24: Detail the method of calculating the outcomes.

#24 1): I don’t understand why the calculation and evaluation of outcomes differentiated by task/quadrant is not possible.

#24-2): Include the correlation considerations you wrote in your answer in the text.

#13: Specify in section 2.3 also the hand tools used for each task/quadrant. In Discussion you mention a mirror (left hand) and handpieces (right hand).

#17: (currently line 179): Replace “… be charged [49].” with “… be charged, in accordance with Ohlendorf et al. [49].

#20: (lines 189-210) Refer only to the original study by McAtamney and Corlett, currently [50]. Remove citations to [47], [51] and [52] in these lines.

#23: (table 3) Improve the table in terms of punctuation and spacing. E.g., “wrist twist score,” (step 4), “were taken ↵ into account”.

#27: If you do not want to summarise the information in a new table, remove the redundancy of the data in both graphs and text. Currently the graphs are useless (they do not add anything to the text).

#28: A time trend would have shown an example of the raw data you processed, and would have helped to better understand the procedure.

#32 + #29 + #34: (results and discussion) The use of the term "ergonomic risk" is confusing, as it is used to comment on both the relative time score and ERP.

#32 + #29 + #37: (results and discussion): You state that the relative time score confirms the results of the median, and ERP confirms the results of median. Are you sure? For example, the wrist has the highest relative time score, whereas the left lower arm has the highest ERP value. I suggested analysing and commenting, only or mainly, ERP. Improve clarity and consistency.

#29: (lines 259-261): You state that the wrist shows the higher score + that the score also (also?) reveals that wrist has the highest risk. Is this not a repetition?

#40 : Is their weight greater than or less than 2 kg? (see step 14 in table 3).

In addition:

#A: Does MSD mean, for you, musculoskeletal disorder or disorders? (see line 17 and 34).

#B (table 2): Move the sentences “All tasks… 3 minutes.” from the caption to the text of the paper.

#C (results): Replace “lower wrist” with “wrist”.

#D (line 337): Check the sentence “… which each …”.

Author Response

The paper has improved. However, I have some comments.

First of all, it is necessary to re-read the text. It is embarrassing for me to suggest deleting the sentence "Klicken oder tippen Sie hier, um Text einzugeben" (lines 60 and 452).

--> We have deleted it, we are sorry about that.

Re-read the whole text, also to correct punctuation. E.g., line 370 “analysisThe present”, line 389 “[13,14].Here”, line 337 “0.68) , which”, line 213 “[50.]”.

-->We have corrected the punctuation throughout the text.

Referring to my previous comments:

#12 + #24: Detail the method of calculating the outcomes.

--> We replaced this sentence: There were 24 recordings stored every second. At the end of each activity, the mean value was calculated from the sum of each measurement recording.

Instead of: At the end of each dental task, the mean value is calculated of all measurement recordings.

#24 1): I don’t understand why the calculation and evaluation of outcomes differentiated by task/quadrant is not possible.

-->The calculation differentiated by quadrants/tasks is possible, but not meaningful to analyse here. Because each quadrant contains a different task, it is not possible to find out whether the ergonomic risk results from the quadrant or from the task. It would be metaphorically comparing apples with oranges. For this, there should be further research with RULA that specifically explores this as a continuation. Our goal in this study, which has a pilot character, was to realistically represent the work of dental students. Therefore, we have chosen these activities. If the participants had performed the same task in each quadrant, this would not be a representative description of the activities taught in dental school.

#24-2): Include the correlation considerations you wrote in your answer in the text.

-->We placed this correlation considerations at the beginning of the results.

#13: Specify in section 2.3 also the hand tools used for each task/quadrant. In Discussion you mention a mirror (left hand) and handpieces (right hand).

--> we have added it in the text.

#17: (currently line 179): Replace “… be charged [49].” with “… be charged, in accordance with Ohlendorf et al. [49].

-->Thank you for pointing this out; we have replaced it.

#20: (lines 189-210) Refer only to the original study by McAtamney and Corlett, currently [50]. Remove citations to [47], [51] and [52] in these lines

-->We have deleted all unnecessary references. However, Vignais is necessary because this research group combined IMU with RULA and our calculations are based on his template.

#23: (table 3) Improve the table in terms of punctuation and spacing. E.g., “wrist twist score,” (step 4), “were taken ↵ into account”.

--> We have corrected the punctuation throughout the table

#27: If you do not want to summarise the information in a new table, remove the redundancy of the data in both graphs and text. Currently the graphs are useless (they do not add anything to the text).

-->We see it differently. While the table shows mean values the figures give a more differentiated look how the mean is composed. They illustrate the percentage of each RULA Score of the entire measurement. The respective ratio of the RULA values, which are shown here relatively, determines the final mean, which is shown in table 4. Accordingly, the outcome values are different.

#28: A time trend would have shown an example of the raw data you processed, and would have helped to better understand the procedure.

-->What do you mean with a “time trend” of raw data? We discussed this point, but are unsure what is meant. The specific procedure of the processing of the raw data is described in our method article “ Maurer-Grubinger, C.; Holzgreve, F.; Fraeulin, L.; Betz, W.; Erbe, C.; Brueggmann, D.; Wanke, E.M.; Nienhaus, A.; Groneberg, D.A.; Ohlendorf, D. Combining Ergonomic Risk Assessment (RULA) with Inertial Motion Cap-ture Technology in Dentistry-Using the Benefits from Two Worlds. Sensors (Basel) 2021, 21, doi:10.3390/s21124077.” cited in the current article.

#32 + #29 + #34: (results and discussion) The use of the term "ergonomic risk" is confusing, as it is used to comment on both the relative time score and ERP.

-->The “relative time score” and ERP are both outcome measures that describe the ergonomic risk. We have specified in appropriate places in the text when an exact outcome variable was meant and not the ergonomic risk per se.

#32 + #29 + #37: (results and discussion): You state that the relative time score confirms the results of the median, and ERP confirms the results of median. Are you sure? For example, the wrist has the highest relative time score, whereas the left lower arm has the highest ERP value. I suggested analysing and commenting, only or mainly, ERP. Improve clarity and consistency.

-->We have improved the clarity and consistency by mainly analysing the ERP (in the results and discussion section).

#29: (lines 259-261): You state that the wrist shows the higher score + that the score also (also?) reveals that wrist has the highest risk. Is this not a repetition?

-->Thank you for pointing this out; we have corrected it.

#40 : Is their weight greater than or less than 2 kg? (see step 14 in table 3).

-->All hand tools weigh much less than 2 kg, it is also mentioned in table 3, step 14.

In addition:

#A: Does MSD mean, for you, musculoskeletal disorder or disorders? (see line 17 and 34).

--> It means disorder, we corrected it in the abstract.

#B (table 2): Move the sentences “All tasks… 3 minutes.” from the caption to the text of the paper.

-->We have moved the sentences from the caption to the text.

#C (results): Replace “lower wrist” with “wrist”.

-->done.

#D (line 337): Check the sentence “… which each …”.

--> We have corrected the sentence; we have deleted the word “each”.

Reviewer 4 Report

The authors' responses to my comments are satisfactory. Improvements have been made in accordance with my recommendations. I noticed small editing errors in the version,  they should be corrected before publication:

  1. L45 - an incorrect form of citations;
  2. Table 2/3/4 - do not comply with mpdi standards (tables should not have darkened rows or columns);
  3. L480-483 - add periods between the first letter of the first name and the first letter of the last name;
  4. reference 8 /L511 - the pages you refer to are missing;
  5. reference 47/L592 - access date missing.

Author Response

  1. L45 - an incorrect form of citations

-->done

  1. Table 2/3/4 - do not comply with mpdi standards (tables should not have darkened rows or columns)

--> We have corrected table 2,3 and 4.

  1. L480-483 - add periods between the first letter of the first name and the first letter of the last name

-->done

  1. reference 8 /L511 - the pages you refer to are missing;

-->done

  1. reference 47/L592 - access date missing.

-->done